# Genetic Characterization and Pathogenesis of Avian Influenza Virus H3N8 Isolated from *Chinese pond heron* in China in 2021

**DOI:** 10.3390/v15020383

**Published:** 2023-01-28

**Authors:** Heng Zhang, Shuyi Han, Bo Wang, Yanan Xing, Guohui Yuan, Ye Wang, Zhilei Zhao, Gaojian Li, Qiaoqiao Li, Jinchao Pan, Wenchao Li, Hongxuan He

**Affiliations:** 1Anhui Province Key Laboratory of Animal Nutritional Regulation and Health, College of Animal Science, Anhui Science and Technology University, Chuzhou 233100, China; 2National Research Center for Wildlife-Borne Diseases, Institute of Zoology, Chinese Academy of Sciences, Beijing 100101, China; 3College of Life Science, University of Chinese Academy of Sciences, Beijing 100101, China; 4College of Agriculture, Ningxia University, Yinchuan 750021, China

**Keywords:** avian influenza virus, H3N8, wild bird, China

## Abstract

In October 2021, a wild bird-origin H3N8 influenza virus-A/Chinese pond heron/Jiangxi 5-1/2021 (H3N8)-was isolated from Chinese pond heron in China. Phylogenetic and molecular analyses were performed to characterize the genetic origin of the H3N8 strain. Phylogenetic analysis revealed that eight gene segments of this avian influenza virus H3N8 belong to Eurasian lineages. HA gene clustered with avian influenza viruses is circulating in poultry in southern China. The NA gene possibly originated from wild ducks in South Korea and has the highest homology (99.3%) with A/Wild duck/South Korea/KNU2020-104/2020 (H3N8), while other internal genes have a complex and wide range of origins. The HA cleavage site is PEKQTR↓GLF with one basic amino acid, Q226 and T228 at HA preferentially bind to the alpha-2,3-linked sialic acid receptor, non-deletion of the stalk region in the NA gene and no mutations at E627K and D701N of the PB2 protein, indicating that isolate A/Chinese pond heron/Jiangxi 5-1/2021 (H3N8) was a typical avian influenza with low pathogenicity. However, there are some mutations that may increase pathogenicity and transmission in mammals, such as N30D, T215A of M1 protein, and P42S of NS1 protein. In animal studies, A/Chinese pond heron/Jiangxi 5-1/2021 (H3N8) replicates inefficiently in the mouse lung and does not adapt well to the mammalian host. Overall, A/Chinese pond heron/Jiangxi 5-1/2021 (H3N8) is a novel wild bird-origin H3N8 influenza virus reassortant from influenza viruses of poultry and wild birds. This wild bird-origin avian influenza virus is associated with wild birds along the East Asian-Australasian flyway. Therefore, surveillance of avian influenza viruses in wild birds should be strengthened to assess their mutation and pandemic risk in advance.

## 1. Introduction

Influenza virus is a single stranded negative-sense RNA virus with a capsule that belongs to the *Orthomyxoviridae* family. Depending on the antigenicity and genetic evolution of its nucleoprotein (NP) and matrix protein (M), it can be divided into four types: A, B, C, and D, of which type A influenza virus is the most harmful. Influenza A virus (IAV) originates from birds and pigs. After infection and transmission in poultry and pigs, IAV gradually acquired the ability to infect humans and caused four pandemics worldwide, including the Spanish H1N1 pandemic in 1918, the Asian H2N2 pandemic in 1957, the Hong Kong H3N2 pandemic in 1968 [1,2], and the Mexican H1N1 pandemic in 2009 [3]. Epidemic occurrence of avian influenza virus (AIV) in horses [4], dogs [5], cats [6], tigers [7], seals [8], whales [9], and other mammals has been demonstrated in previous studies. Wild birds are considered natural reservoirs of AIV [10], with *Anseriformes* (mainly ducks, swans, and geese) and *Charadriiformes* (mainly gulls, terns, and waders) playing an important role in the epidemic and transmission of AIV [11]. AIV can be transmitted to poultry by wild birds, causing severe social and economic losses, and occasionally to humans, causing zoonoses.

The subtypes of influenza viruses are diverse, and the structure of the viral genome and the specific functions of its proteins result in frequent antigenic variation [12]. Antigen drift and antigen switching are two key processes in the evolution of influenza viruses. Antigenic drift on HA proteins can generate new strains that escape pre-existing antibody immunity [13] and is the main reason for the annual update of influenza vaccines to prevent seasonal influenza [14]. Antigen switching is a sudden and drastic change in influenza virus antigen, which is a qualitative change in antigenicity [15]. Different antigenic strains infecting the same cell can reassort genomic fragments, resulting in hybrid offspring. Gene reassortment is very common in AIV, resulting in a wide diversity of influenza viruses in birds. Antigenic drift and antigenic switching are important reasons why influenza virus continues to circulate worldwide and is difficult to prevent and control.

AIVs can be divided into highly pathogenic avian influenza viruses (HPAIVs) and low pathogenic avian influenza viruses (LPAIVs) based on their differential pathogenicity to chickens. Most AIVs are LPAIVs, and the most common HPAIVs subtypes include HPAIVs H5N1 and HPAIVs H7N9. HPAIVs can cause severe respiratory diseases or a large number of deaths, while LPAIVs are usually asymptomatic or cause mild upper respiratory illness [16]. Although the damage caused by LPAIVs is not as great as that caused by HPAIVs, they also play an important role in the spread and mutation of avian influenza viruses. For example, H9N2 LPAIVs provides internal genes for H5N1, H7N9, H5N6, and other HPAIVs [17,18,19]. H3N8 AIVs are one of the most commonly found subtypes in wild birds and poultry. It is worth noting that H3N8 influenza virus can bind to both the α2,3-sialic acid (SA) of avian influenza virus and the α2,6-SA of human avian influenza virus. In addition, H3N8 influenza virus has a wide host range that can infect not only birds but also a variety of mammals, such as horses [20], dogs [21], pigs [22], cats [23], seals [24], camels [25], and donkeys [26], etc. In April 2022, the first human infection with H3N8 AIV was reported in Henan, China [27]. H3N8 AIV breaks the interspecies barrier and spreads to humans, further increasing the epidemic risk in mammals and humans [28]. Seasonal migration of wild birds has promoted the global spread of AIVs. Therefore, strengthening the surveillance of AIVs in wild birds is very important to prevent and control the spread of AIVs in wild birds to poultry.

During surveillance of wild bird AIVs in Suichuan, Jiangxi Province, a novel wild bird origin H3N8 AIV was isolated from *Chinese pond heron* in October 2021. Since there are few studies on the transmission mechanism of H3N8 AIVs in wild birds, the aim of this study is to understand the source and transmission risk of the wild bird origin H3N8 AIV.

## 2. Materials and Methods

### 2.1. Samples Collection and Virus Isolation

On 27 October 2021, 126 oropharyngeal and cloacal swabs were collected from wild birds during AIV surveillance in Suichuan, Jiangxi Province, China. Swab samples were placed into 1.5 mL Eppendorf tubes with 1 mL PBS buffer containing antibiotics (penicillin 2000 U/mL and streptomycin 2000 U/mL) and then stored and transported on ice.

The 0.2 mL supernatants (double antibody treatment overnight) of the positive avian influenza samples were inoculated into the allantoic cavities of 10-day-old specific-pathogen-free (SPF) embryonated chicken eggs (Boehringer Ingelheim, Beijing, China). The egg was incubated at 37 °C and then chilled at 4 °C overnight after death or 72 h. Allantoic fluid was harvested, and hemagglutinin activity was determined using 1% red chicken blood cells.

### 2.2. RNA Extraction and RT-PCR

Total RNA was extracted from hemagglutinin-active positive allantoic fluid using TRIzol Reagent (Invitrogen) and reverse transcribed using primer Uni12 5′-AGCRAAAGCAGG-3′ and GoScript™ Reverse Transcriptase System (Promega, Madison, WI, USA). PCR amplification was used to subtype hemagglutinin (HA) and neuraminidase (NA), and all eight segments of the virus were amplified by RT-PCR using the universal primer set (Appendix A).

### 2.3. DNA Cloning and Gene Sequencing

All RT-PCR products were purified using the FastPure^®^ Gel DNA Extraction Mini Kit (Vazyme, Nanjing, China). The purified PCR products were cloned into the pCE2 TA/Blunt-Zero vector (Vazyme, Nanjing, China) and transformed into Fast-T1 competent cells. The recombinant plasmids were screened on Luria–Bertani (LB) agar plates containing ampicillin (1 μL/mL). Positive clone bacterial fluids were identified using 2× Rapid Taq Master Mix and M13 primer according to the manufacturer’s instructions. Bacterial fluid conforming to eight influenza virus gene fragments were sent for sequencing (BGI, Beijing, China).

### 2.4. Genetic and Phylogenetic Analysis

The genome sequences of A/Chinese pond heron/Jiangxi 5-1/2021 (H3N8) were obtained by sequencing results, and closely related sequences were downloaded from BLAST searches against GISAID and GenBank. MEGA5 and the reserved CDS region were used to align all segmented sequence datasets. A nucleotide substitution model was used to estimate the best fit of eight genes using jModeltest2 [29]. An uncorrelated relax-clock Bayesian Markov chain Monte Carlo method in BEAST v1.10.4 [30] was used to estimate divergence times and rates of nucleotide substitutions. To determine which phylodynamic models fit best, we performed different combinations of relaxed-clock models (i.e., exponential and lognormal models) and branch rate models (i.e., constant size, exponential growth, Bayersian SkyGrid, and GMRF Bayersian SkyGrid models). To achieve convergence, the MCMC chain was run for 500,000,000 iterations, with sampling every 50,000 steps. Tracer v1.6.0 was used to evaluate the model comparison analyzes (AICM analysis; [31] and sufficient sampling from the posterior (effective sample size 200)). Tree Annotator v1.10.4 generated and summarized a maximum clade credibility (MCC) tree with a 10% burn-in. The entire phylogenetic tree was visualized using FigTree v1.4.4.

### 2.5. Determination of 50% Egg Infectious Dose (EID_50_) and 50% Tissue Culture Infectious Dose (TCID_50_)

To determine EID_50_, serial 10-fold dilutions of the viruses were inoculated to 10-day-old embryonated SPF chicken eggs with 100 μL, four eggs for each dilution. The eggs were then incubated at 37 °C for 72 h, and the EID_50_ of the harvested allantoic fluids was determined using the method of Reed and Muench [32]. To determine the TCID_50_ titer, Darby Canine Kidney (MDCK) cells were cultured in 96-well flat-bottomed plates. According to the standard operating procedures (SOP) of the National Influenza Center of China, the virus allantoic fluid was semi logarithmically diluted with the virus culture medium containing 2 μg/mL TPCK-trypsin and then inoculated into 96-well flat-bottomed plates while 90% MDCK cells confluence, with each dilution of 4 wells. The cell culture plates are incubated at 37 °C and 5% CO_2_ for 1 h. After incubation, the virus allantoic fluid is removed and the plate is washed twice, then 100 μL of virus culture medium containing 2 µg/mL TPCK-trypsin is added to the 96-well plates. Virus-infected cells were incubated at 37 °C and 5% CO_2_ for 72 h, and TCID_50_ titers were calculated using Reed and Muench methods.

### 2.6. Animal Experiment

To evaluate the adaptability of the virus to mammals, BALB/c mice aged 6–8 weeks (SiPeiFu, Beijing, China) (n = 11) were intranasally infected with 10^6^ EID_50_/mL H3N8 avian influenza virus 50μL, and the control group was inoculated with the same amount of PBS; body weight and survival rate were monitored for 14 dpi. Afterwards, the mice were euthanized on 3 dpi, 5 dpi, and 14 dpi, and lung and brain tissues were collected. TCID_50_ of lung and brain tissues were measured to detect virus titers in the homogenate supernatant [33].

## 3. Results

### 3.1. Virus Isolation and Homology Comparison

During surveillance of avian influenza virus in wild birds in Suichuan, Jiangxi Province, the primer used to identify the M gene of avian influenza virus found that the sample numbered JX 10-27 5-1 was positive (Appendix A). The strain was successfully isolated from 10-day-old SPF chicken embryos. According to the influenza virus subtype identification primer, the result of its surface glycoprotein HA H3, and NA was N8. It was designated as A/Chinese pond heron/Jiangxi 5-1/2021 (H3N8) (JX 5-1).

The full-length sequences of eight genes of JX 5-1 were obtained by monoclonal plasmid, and the homology of all eight gene segments of JX 5-1 was compared in GenBank (Table 1 and Appendix A). From the Table 1, it can be concluded that JX 5-1 is the recombination of different AIV subtypes in Asian poultry and wild waterfowl. The HA gene of JX 5-1 was close to A/chicken/Guangxi/165C7/2014 (H3N2) with an identity of 95.36%. The most closely related virus of the NA gene of JX 5-1 was A/Wild duck/South Korea/KNU2020-104/2020 (H3N8) with an identity of 99.30%. The internal gene (PB2) of JX 5-1 showed a close relationship with A/duck/Tottori/311215/2020 (H5N2), with 98.96% nucleotide identity; other internal genes (PB1, PA, NP, M, and NS) were similar to those of isolated AIV strains from China.

### 3.2. Phylogenetic Analysis and Hypothesis for Reassortment Event of Each Gene Segment

To investigate the origin of virus A/Chinese pond heron/Jiangxi 5-1/2021 (H3N8) and the genetic relationships of internal genes to domestic poultry and wild birds in China and neighboring countries, we performed the phylogenetic tree of each segment using the closest sequence downloaded from GISAID and GenBank. All genes of JX 5-1 belong to the Eurasian lineage according to the phylogenetic analysis (Figure 1 and Appendix A). From the above homology and phylogenetic analysis, the source locations of the JX5-1 internal gene are presumed to be in China, Korea, and Japan (Figure 2). Evolutionary reassortment tracking analysis shows that HA and M genes of JX 5-1 are closely associated with AIVs H3N2 and H7N7 in chickens and ducks in southern China. The origin of the NS and NP genes is similar to that of the HA and M genes; the difference being that the NS and NP genes are likely H3N8 and H7N3 recombined in mallard ducks (Figure 3).

The NA gene reassortment may have occurred in Korea before being transmitted to China by wild ducks. The backbone of PB2 and PB1 genes might be due to the reassortment of Korea isolates (H7) which are privileged in South Korea; likewise the PA gene might have been transmitted by the Korean mallard. Then, probably in 2019, the PB1 and PA genes were transmitted to China by wild duck migration and reassortment with the AIVs in Chinese ducks. After the PB2 gene was transmitted to Vietnam through waterfowl ducks, it might have reassorted with the AIVs in Japanese ducks (Figure 3). In summary, JX-5-1 is a multiple recombinant strain of several avian influenza viruses found in migratory waterfowl and local poultry.

The evolutionary rate of eight gene segments of JX 5-1 was estimated using Bayesian analysis (Appendix A). Among the eight genes, the evolution rate of the genes NP and PA was significantly faster than that of the other genes. HA gene has the slowest evolutionary rate compared with the other genes. The effective population size of JX 5-1 was estimated based on Bayesian phylodynamics and the Ne value (number of genes that effectively produce the next generation) (Figure 4). The Ne value of HA shows a decreasing trend after 2014 while that of NA plateaus, thus inferring that there was no pandemic outbreak of H3N8 subtype avian influenza virus after 2014.

### 3.3. Molecular Analysis

We examined the molecular properties of amino acid sequences to assess the risk of JX 5-1 to mammals. The results show that the amino acid sequence motif at the cleavage site of the HA protein is PEKQTR↓GLF with one basic amino acid, which is characteristic of low pathogenic AIV. Q226 and T228 of the receptor-binding site on HA have the characteristics of AIV preferentially binding to the alpha-2,3-linked sialic acid (SA α- 2, 3-Gal) receptor (Table 2). There were no mutations at E627K and D701N of PB2 protein, which may increase mammalian adaptability [34]. However, mutations L89V, G309D, T339K, and I495V of the PB2 protein may increase polymerase activity in mouse cells (Table 3).

In addition, there are N30D and T215A mutations in M1 protein of JX5-1 and P42S mutations in NS1 protein. These mutations have been reported to increase the virulence of H5N1 avian influenza virus in mammals [35,36]. There are also mutations in PB1 (L473V) and PA (L295P, N383D, M423I, V476A, and V630E) proteins that have been proven in previous studies to be some of the mutations that enhance adaptation in mammals [37,38,39,40]. These findings indicate that JX 5-1 still preferentially binds to avian receptors, some internal gene has acquired mutations that may increase the virulence and transmission in mammalian hosts.

**Table 3 viruses-15-00383-t003:** Summary of data obtained from the mutational analysis of eight genes from AIVs of multiple avian species with the H3N8 (JX 5-1) isolate. (“-”—no amino acid was found).

Viral Protein	Amino Acid	JX 5-1	HN-410	South Korea2020 (H3N8)	Zhejiang2013 (H3N8)	Xuyi2014 (H3N8)	Amur Region2020 (H3N8)	Comments	Reference
PB2	L89V	V	V	V	V	V	V	Increased polymerase activity and virulence in mammals	[41]
	G309D	D	D	D	D	D	N	Increased polymerase activity and virulence in mammals	[41]
	T339K	K	K	K	K	K	T	Increased polymerase activity and virulence in mammals	[41]
	E627K	E	K	E	E	E	E	Mammalian host adaptation	[42,43]
PB1	H436Y	Y	Y	Y	Y	Y	Y	Increased polymerase activity and virulence	[44]
	L473V	V	V	V	V	V	V	Increased polymerase activity and replication efficiency	[45]
PA	K356R	K	R	K	K	K	K	Enhanced virulence and mammalian adaptation	[46]
	N383D	D	D	D	D	D	D	Increased polymerase activity and mammalian adaptation.	[37]
	N409S	S	N	S	S	S	S	Increased polymerase activity, viral replication and virulence to mammalian	[47]
M1	V15I	V	I	V	V	V	V	Increased virulence in mammals	[48]
	N30D	D	D	D	D	D	D	Increase pathogenicity and transmission in mammals	[49]
	A166V	V	A	V	V	V	V	Increased polymerase activity and virulence in mammals	[50]
	T215A	A	A	A	A	A	A	Increased virulence in mammals	[35]
M2	V27I	V	V	V	V	I	V	Reduce the sensitivity of Adamantane	[51]
	S31N	S	N	S	S	S	S	Reduce the sensitivity of Adamantane	[52,53]
	L55F	L	F	L	L	L	L	Increased transmission	[54]
NS1	P42S	S	S	S	S	-	S	Enhanced virulence in mice	[36]

### 3.4. Pathogenicity in Mice

To evaluate the pathogenicity of JX 5-1 in mammals, we inoculated 6-week-old BALB/c female mice with 50 μL of 10^6^EID_50_ virus (Appendix A). During the observation of clinical signs, ruffled fur, depression, and dyspnea were not particularly evident, but their activity was attenuated compared to the control group. The body weight of the infected mice decreased transiently, and gradually returned to normal after the body weight decreased to the lowest point at three days post infection (dpi) (Figure 5A). The autopsy results showed that on the 14 dpi, there were obvious lesions in the lung tissue of the mice, accompanied by intestinal edema (Appendix A). To detect the expression of influenza virus in mice, the viral titer in the lungs and brain at 3 dpi, 5 dpi, and 14 dpi was determined by TCID_50_. The results showed that the replication efficiency of JX5-1 was low in the lung and brain of mice at 3 dpi and 5 dpi, and no virus was detected at 14 dpi. (Figure 5, Appendix A). This indicates that JX5-1 has low pathogenicity to mammals and is not well adapted to mammalian hosts.

## 4. Discussion

Wild birds are carriers of AIVs, which usually have little or no pathogenicity. In addition, infection with LPAIVs may not affect the movements of mallards, allowing the virus to spread along the migration route [55]. Jiangxi Province is located on the East Asian-Australian migratory bird flyways, and a large number of migratory birds pass through the region each year. During surveillance of AIVs in wild birds in Suichuan, Jiangxi Province, we found an avian influenza virus subtype H3N8.

In this study, we analyzed this H3N8 subtype avian influenza virus isolated from wild birds. Phylogenetic analysis revealed that JX 5-1 is a reassortant virus of Eurasian lineage. The H3 subtype of JX 5-1 has the highest homology with the H3 subtype prevalent in ducks and chickens in China, and the N8 subtype was closely related to H3N8 AIVs in wild ducks in Korea. This means that wild birds carrying the avian influenza virus spread the N8 subtype during migration from Korea to China and recombined with the H3 subtype in local poultry. Phylogenetic analysis of a wild bird-origin H3N8 AIV found in Xinjiang showed that its N8 originated in Mongolia and was also associated with wild bird migration [56]. Although the N8 gene of the H3N8 AIV infecting humans is also related to migratory birds, it belongs to the North American lineage rather than the Eurasian lineage [28]. Molecular epidemiological studies in domestic poultry in southern China revealed that reassortment between the Eurasian lineage and North American lineage is common in H3Ny subtypes [57].

The genome-wide analysis of JX 5-1 reveals that its internal genes have a diverse variety of origins and that many of them are clustered with AIVs in Asian nations near China during the course of genetic evolution. Two H3N8 influenza viruses with wild bird origins—XJ47 and GZ—were shown to have comparable internal gene source dynamics to JX5-1 in a different investigation [58]. It is speculated that the migration of wild birds among Asian countries caused this frequent gene exchange. The H7N9 avian influenza virus was the first to infect people in 2013 [19]. Its HA gene was introduced from H7 among migratory birds to poultry, and its NA gene was closely connected to wild birds in Korea [19]. The H10 and N8 genes of the H10N8 avian influenza virus that infects humans may have originated through the recombination of several influenza viruses in wild birds; following infection of poultry, H9N2 gives them internal genes and gains the capacity to infect people [59]. Therefore, wild birds have a significant influence on how the influenza virus develops, disseminates, and is transmitted to poultry and mammals [60]. The global spread of the influenza virus is aided by wild bird migration. China is traversed by four of the nine migratory flyways: the West Asian-East African flyway, Central Asian flyway, East Asian-Australasian flyway, and West Pacific flyway. The avian influenza virus will be spread by wild birds that are bringing it to the nations along the migration flyways. Domestic poultry and wild birds may reassort and exchange genes, creating new reassortant strains or adaptive mutations [61,62,63,64].

The receptor binding characteristic of the influenza virus is that avian influenza virus preferentially binds SA α- 2, 3-Gal receptor, human influenza virus preferentially binds alpha-2,6-linked sialic acid (SA α- 2,6-Gal) receptor [65,66]. The ability of a virus to adapt to new hosts can be improved by the process of changing the binding properties of its receptors from preferentially binding SA α- 2, 3-Gal receptor to SA α- 2,6-Gal receptor. The H3N2 avian influenza virus’s HA gene underwent mutations Q226L and G228S in 1968, which made the virus preferentially attach to the SA α- 2,6-Gal receptor and led to the epidemic in Hong Kong [67]. For H2 and H3 viruses, the substitution of amino acid sites Q226L and G228S will affect the receptor binding specificity of HA [68]. The results of the molecular study of JX 5-1 revealed that Q226 and T228 on HA are the receptor-binding sites that still preferentially bind to avian receptors. According to reports, the PB2 proteins 627K and 701D can boost polymerase activity and improve pathogenicity to mammals, which are crucial molecular indicators for the avian influenza virus to adapt to mammalian hosts [42,69,70]. Mutations in the wild bird-origin H3N8 avian influenza virus PB1 protein S524G also enhance virulence and fitness for mammalian transmission in a recent study [71]. In this study, JX 5-1 did not have mutations at these sites. Nevertheless, mutations in N30D of M1 protein, T215A and P42S of NS protein may enhance pathogenicity and mammalian transmission [35,48,54]. The results of animal experiments showed low pathogenicity of JX5-1 in mice and inefficient replication of the virus in the lungs, indicating that the strain is not well adapted to mammals. In another study, both strains of wild bird-origin H3N8 avian influenza virus were able to replicate efficiently in mice and guinea pigs [58].

The H3N8 AIVs have been repeatedly detected in wild birds and poultry in China, particularly in ducks [72,73,74,75,76]. H3N8 avian influenza cross-species transmission cases have been documented in the past for a variety of animal species, including equines and seals [20,77]. Previous research has shown that H3N8 AIVs isolated from seals can spread through respiratory droplets in ferrets and replicate successfully in human lung cells [78]. The first human case of H3N8 avian influenza virus infection was reported in China on 10 April 2022 [27]. The patient was a young child who had come into contact with poultry before becoming ill, and it was thought that hens raised at home may have been the source of the infection. After that, a child who had previously been exposed to live poultry was also reported to have the H3N8 avian influenza virus in Changsha [57]. Despite the fact that the H3N8 avian influenza virus passed from poultry to people by accident, it is important to note that H3N8 influenza viruses are highly susceptible to recombination and the source of internal genes is complicated, which increases the potential for a pandemic.

In conclusion, the JX 5-1 is a novel reassortment H3N8 influenza virus with wild bird origin. All of its surface genes, including H3 related to Chinese poultry and N8 related to Korean wild ducks, are of Eurasian lineage. Internal genes are a reassortment of multiple subtypes of avian influenza viruses. Although an assessment of the effective population size of H3N8 subtype avian influenza viruses suggests that there have been no outbreaks in recent years, there is still a need to constantly monitor the risk of a pandemic and to increase the surveillance of the H3N8 avian influenza virus in wild birds, particularly along migration flyways where wild birds congregate in high numbers.

## Figures and Tables

**Figure 1 viruses-15-00383-f001:**
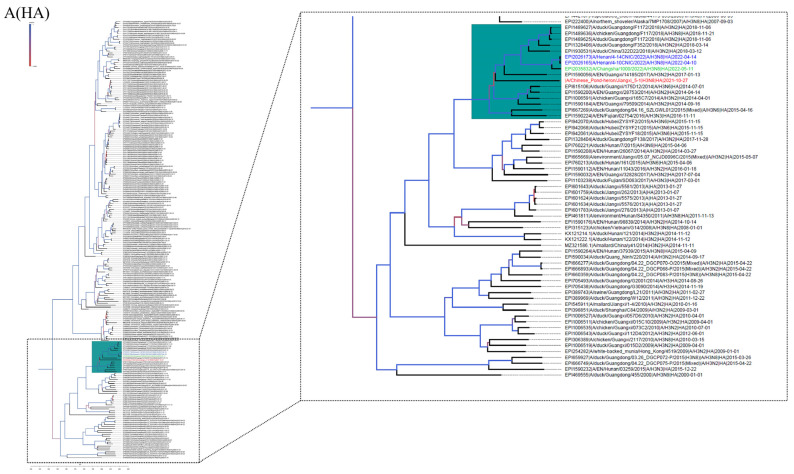
Phylogenetic analysis of HA (**A**), NA (**B**) of JX 5-1. The phylogenetic trees were constructed using gene sequences identified in NCBI or GISAID Blast analyses. JX 5-1 is marked in red, the closest related sequence is marked in blue. The trees were built using BEAST (v1.8.4) and illustrated using FigTree (v1.4.2).

**Figure 2 viruses-15-00383-f002:**
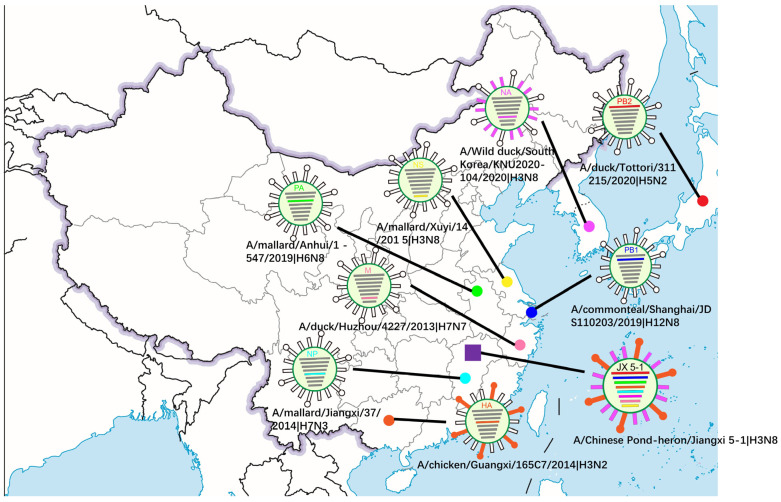
Locations of the putative origin of genomic compositions of the H3N8 (JX 5-1).

**Figure 3 viruses-15-00383-f003:**
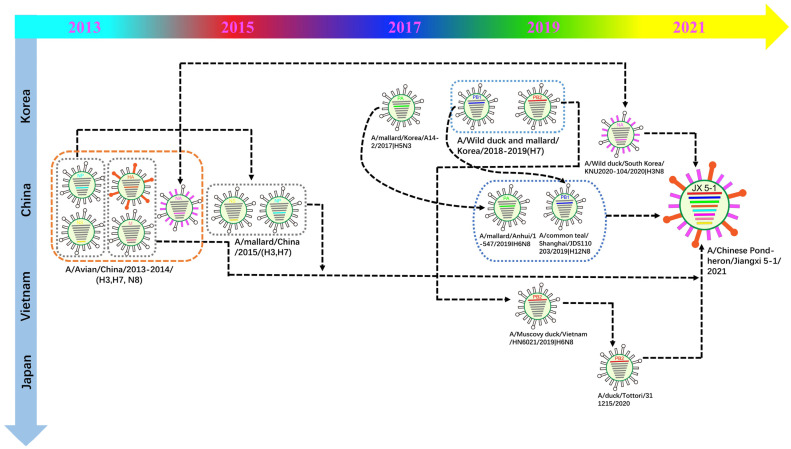
Original reassortment events of the novel avian influenza isolate H3N8 (JX 5-1).

**Figure 4 viruses-15-00383-f004:**
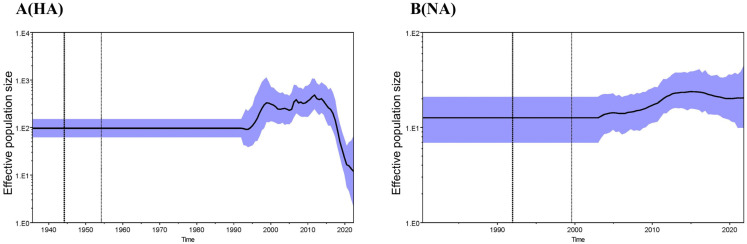
Effective population size of HA and NA. Estimation of effective population size of HA and NA genes of JX 5-1 using the Bayesian SkyGrid model. Panels (**A**,**B**) represent the HA and NA genes, respectively.

**Figure 5 viruses-15-00383-f005:**
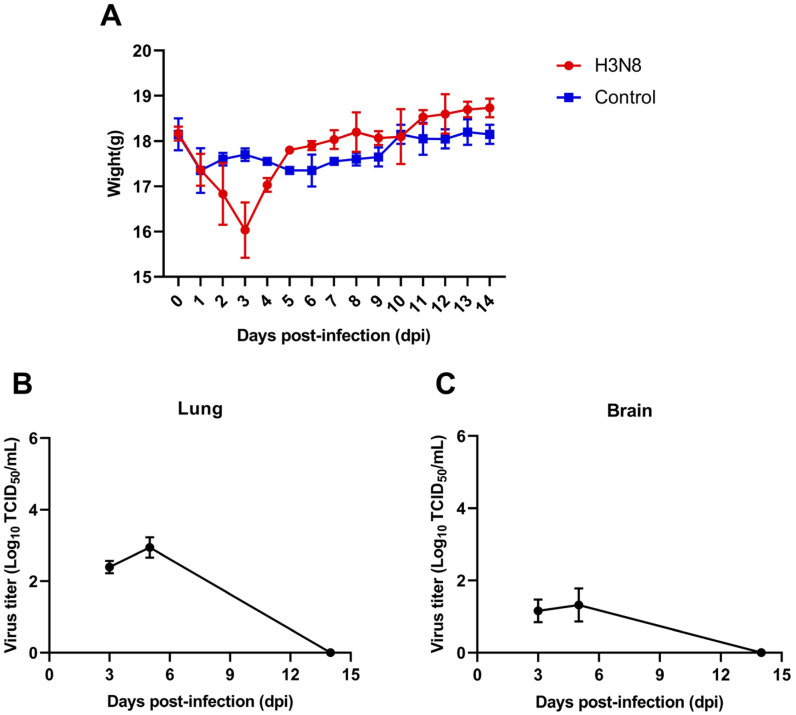
Pathogenicity of the H3N8 (JX 5-1) isolate in vivo. For each virus strain, BALB/c mice were intranasally infected with 10^6^ EID_50_/mouse concentrations of the virus. (**A**) Mean body weight (n = 3), (**B**) virus titer in the lung at 3 dpi, 5 dpi and 14 dpi (*n* = 3), (**C**) virus titer in the brain at 3 dpi, 5 dpi and 14 dpi.

**Table 1 viruses-15-00383-t001:** Sequence identity of each gene between the JX 5-1 virus and the closest homologs in the GenBank database.

Gene	Viruses with Greatest Homology	Accession	Identity (%)
PB2	A/duck/Tottori/311215/2020 (H5N2)	LC656330.1	98.96%
PB1	A/common teal/Shanghai/JDS110203/ 2019 (H12N8)	MN795765.1	99.43%
PA	A/wild goose/dongting lake/121/ 2018 (H6N2)	MH727479.1	98.68%
HA	A/chicken/Guangxi/165C7/2014 (H3N2)	KT022317.1	95.36%
NP	A/canine/Zhejiang/S34/2015 (H3N8)	MH018583.1	97.33%
NA	A/Wild duck/South Korea/KNU2020-104/2020 (H3N8)	OK236005.1	99.30%
M	A/duck/Huzhou/4227/2013 (H7N7)	KP413918.1	98.37%
NS	A/duck/China/F1473-2/2016 (H6N2)	MT828327.1	98.57%

**Table 2 viruses-15-00383-t002:** Comparison of the hemagglutinin (HA) receptor-binding sites and neuraminidase (NA) gene segments of the novel avian H3N8 isolate and those of high related avian H3N8 isolates.

Virus Strain	HA Receptor-Binding Residues (H3 Numbering)								NA
	Cleavage Sites	135	138	160	186	192	226	228	Stalk Region Deletion
JX 5-1	PEKQTR↓GLF	E	T	A	N	R	Q	T	No deletion
HN-410	PEKQTR↓GLF	D	T	A	N	K	Q	T	No deletion
South Korea2020 (H3N8)	PEKQTR↓GLF	E	T	A	N	K	Q	T	No deletion
Zhejiang2013 (H3N8)	PEKQTR↓GLF	E	T	A	N	K	Q	T	No deletion
Xuyi2014 (H3N8)	PEKQTR↓GLF	E	T	D	N	K	Q	T	No deletion
Amur region2020 (H3N8)	PEKQTR↓GLF	E	T	A	N	K	Q	T	No deletion

## Data Availability

All the data presented in this study are available in this article and Appendix A.

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
