# Peer review of "Genetic Characterization and Pathogenesis of Avian Influenza Virus H3N8 Isolated from Chinese pond heron in China in 2021"

_viruses, 2023, doi:10.3390/v15020383_

Round 1

Reviewer 1 Report

Zhang et al. conducted the comprehensive characterization of H3N8 low pathogenicity avian influenza virus isolated from a wild bird in China. Although the authors could make great efforts for this, there are less novelty in this manuscript. The authors must re-arrange the data and discussions to publish this article in Viruses.

<Major points>

Most critical point in this article is that the authors did not obtain the obvious data of JX 5-1 infection in mice. Though the weight loss was confirmed after virus challenge (but at worse 85% and recovered in three days), there were no clinical signs which the authors wrote down. Moreover, the virus titers in lungs are so low; I do not deny the possibility that the high dose of challenge viruses were just remaining in the lumen of lungs. If the infectious virus replicates in lungs, the titer of virus is usually more than 10^3/ml and more in 5dpi (see your reference No. 40). To make clear this uncertainty, the authors must check the antibody response of mice survived for 14 days after challenge or titrate the infectious virus (not measuring the genome by RT-PCR) in 3 and 5 dpi. Otherwise, the authors cannot conclude JX 5-1 could replicate in mouse lung.

The authors must indicate the virus titer of the brain in the main text and argue the possibility of systemic infection of JX 5-1. This must be a very important finding.

The results of phylogenetic analysis could not indicate the parental viruses of the reassortant virus, but indicate the closest ancestor(s) ever reported. Throughout the manuscript, the generation of the reassortant viruses were assertively determined only using the available data without focusing on undetected viruses. The authors must modify the style of description.

Evaluation of the pathogenicity of JX 5-1 was conducted in mouse model and compare the amino acids which potentially affect on the pathogenicity in mouse. However, most of the molecular markers which the authors indicated in the manuscript were originally derived from H5 high pathogenicity avian influenza viruses (see reference No. 35, 40). Are these markers also affecting on the pathogenicity of low pathogenicity avian influenza virus infection? This is coming from the clear difference of pathogenesis in mouse after infection with low and high pathogenicity avian influenza viruses.

Overall, it is over discussed. Relating to my previous criticism, phylogenetic analysis just indicates the closest relationship between the isolate and strains ever reported. It did not surely determine the ancestor of the isolate. Because the natural reservoir of AIVs are water birds, many of the viruses had been sustained that population with causing gene reassortment. The authors argued the potential pathogenicity of AIV infection in human or other mammalians referring the results of only mouse experiment.

I am so confused in the novelty of this manuscript. The authors also pointed out in Line 328-329, “wild bird-origin H3N8 AIVs can potentially adapt well to a mammalian host”.  What is the novelty of this manuscript beyond the reference paper of No. 74?

Overall, there are several typos and grammatical errors.

<Minor points>

L27-28: Importance of these amino acid mutations seem to be revealed in this manuscript. In this paper, already know amino acid mutations were just confirmed in the isolated virus. Please make clear this point not to make audience confused.

L28-30: This content (“replicate effectively”) is not matching to the content in Line 259 (“it will not have serious effects”).

L79-80: The authors must indicate the reference which prove the infection of H3N8 AIV from birds to human.

Figure 1. These trees are too small to be seen. Not all of the trees are necessary; except the HA tree, others should be put in Supplemental data. The HA tress should be enlarged in the main manuscript.

Figure 2. I strongly recommend that the authors should remove the Figure 2 because there are no descriptions regarding the Figure 2. If the authors would like to keep this, add the necessary information about Figure 2 in the main text, and indicate the flyways of bird migration in the map so as to understand the occurrence of reassortment in this region.

L213-219; I assume this part is also relating to Figure 4, but there are no indications. Furthermore, there are no Discussion regarding these results. Unless describing the discussion and consideration in your end, these results and figures are just indication without any intention of the authors, meaning that this figure is not worthy being kept in here.

L232-233: Were these molecular markers confirmed in mouse “cells”?

Figure S1: The author must add the figure legend of the photos of autopsy.

Figure 5B: There are no error bars; is this the results of only one mouse?

L298-303: There are no references to assist these contents.

L320-323: Again, I cannot agree the content that “JX 5-1 can be effectively replicated in”. Please rephrase it or indicate more obvious evidence to mention it.

Reviewer 2 Report

Comments to the Author

The manuscript entitled " Genetic Characterization and Pathogenesis of Avian Influenza

Virus H3N8 Isolated from Chinese pond heron in China in 2021" represents a considerable amount of work. The following comments need to be addressed before the manuscript is suitable for publication in viruses Journal.

-          Line 147: On what basis did the authors choose the infective dose?

-          Line 149: Why did authors choose these days for examination?

-          In the discussion part, why authors did not discuss the results of pathogenicity in mice?

Round 2

Reviewer 1 Report

Thanks for well improvment.